# Overweight, Obesity, and Associated Risk Factors among Students at the Faculty of Medicine, Jazan University

**DOI:** 10.3390/medicina60060940

**Published:** 2024-06-04

**Authors:** Sameer Alqassimi, Erwa Elmakki, Areej Siddiq Areeshi, Amani Baker Mohammed Aburasain, Aisha Hassan Majrabi, Enas Mohammed Ali Masmali, Eman Adel Ibrahim Refaei, Raghad Abdu Ali Mobaraki, Reem Mohammed A. Qahtani, Omar Oraibi, Majid Darraj, Mohammed Ali Madkhali, Mostafa Mohrag

**Affiliations:** 1Department of Internal Medicine, Faculty of Medicine, Jazan University, Jazan P.O. Box 114, Saudi Arabia; eelmakki@jazanu.edu.sa (E.E.); ooraibi@jazanu.edu.sa (O.O.); mdarraj@jazanu.edu.sa (M.D.); mmedkhali@jazanu.edu.sa (M.A.M.); mmohrag@jazanu.edu.sa (M.M.); 2Faculty of Medicine, Jazan University, Jazan P.O. Box 114, Saudi Arabia; areshiareej@gmail.com (A.S.A.); amaniaburasain@gmail.com (A.B.M.A.); dr.aisha1914@gmail.com (A.H.M.); enas.m.masmali@gmail.com (E.M.A.M.); emanrefay1111@gmail.com (E.A.I.R.); iraghad89@icloud.com (R.A.A.M.); reem01775@gmail.com (R.M.A.Q.)

**Keywords:** overweight, obesity, prevalence, risk factors, medical students, Jazan University, physical activity, dietary habits

## Abstract

*Background and Objectives:* This study aimed to determine the prevalence of overweight, obesity, and the associated risk factors among medical students at Jazan University in Saudi Arabia. *Materials and Methods:* A cross-sectional study was conducted among 228 medical students from their second to sixth academic years at the Faculty of Medicine, Jazan University. A self-administered questionnaire was used to collect data regarding sociodemographic characteristics, physical activity, dietary habits, comorbidities, medication use, family history, and lifestyle factors. Anthropometric measurements including height, weight, and waist circumference were recorded. Chi-square test and binary logistic regression were used to identify the risk factors associated with obesity. *Results:* The prevalence of overweight and obesity among the participants was 13.3% and 15%, respectively. Hence, the combined prevalence of overweight and obesity is 28.3%. The mean weight was 63.39 ± 18.93 kg, and the mean height was 163.48 ± 9.78 cm. On the other hand, 17.3% of participants were underweight, whereas 54.4% had normal BMI. Most of the participants (61%) did not engage in regular exercise. A high proportion consumed fruits (82.9%) and vegetables (58.8%) 3 or fewer days per week, and 84.2% consumed 3 or fewer meals per day. Fast-food consumption more than 3 days per week was reported by 42.1% of participants. Obesity was not significantly associated with sociodemographic factors, physical activity, dietary habits, comorbidities, medication use, or family histories. However, those with a monthly family income of SAR 15,000–24,999 had significantly lower odds of obesity than those in the lowest income group (OR 0.230, *p* = 0.045). *Conclusions:* The prevalence of overweight and obesity among medical students at Jazan University is high. Although no significant associations were found between obesity and most risk factors, this study highlights the need for interventions that promote healthy lifestyles among medical students. Further research is needed to identify effective strategies for preventing and managing obesity in this population.

## 1. Introduction

Overweight and obesity are complex, multifaceted chronic conditions that have become global health crises [1,2,3].

The World Health Organization (WHO) defines overweight and obesity as excessive accumulation of body fat that poses health risks [4]. The recommended tool for categorizing obesity is the body mass index (BMI), with a BMI between 25 and 30 kg/m^2^ indicating overweight, and a BMI above 30 kg/m^2^ indicating obesity [5,6]. 

The global prevalence of obesity is escalating at an alarming rate, with projections suggesting that, by 2030, up to 57.8% of the world’s adult population could be either overweight or obese [4,5].

University years are a particularly challenging period for obesity, as the transition from high school often involves changes in physical activity and eating habits that can lead to weight gain [7]. Research in the Arab region has shown that rapid socio-cultural changes due to economic growth have significantly impacted dietary patterns, contributing to the rising rates of overweight and obesity in countries such as Saudi Arabia [8]. Additionally, psychological stress associated with college life, especially in medical schools, has been identified as another critical factor in the development of obesity [9,10]. 

Obesity is a complex issue with multiple contributing factors, including diet, genetics, physical activity, physiology, and behavior [10,11]. Healthcare professionals are expected to be role models for healthy living, but studies have shown that obesity is a problem among medical students and healthcare workers in many countries [12,13].

Studies have demonstrated a high prevalence of obesity among university students. Peltzer et al. found that the overall prevalence of overweight and obesity among university students in low- and middle-income countries is 22.4% and 5.0%, respectively [14]. Another study conducted in Saudi Arabia by Al-Rethaiaa et al. reported that the prevalence of overweight and obesity among male university students was 21.8% and 15.7%, respectively [8].

Several risk factors have been associated with obesity in university students. Al-Nakeeb et al. found that physical inactivity, sedentary behavior, and unhealthy dietary habits were significant predictors of overweight and obesity among university students in the United Arab Emirates [15]. Similarly, Musaiger et al. reported that a lack of physical activity, consumption of fast foods, and prolonged screen time were associated with obesity among university students in Kuwait [16].

Medical students are a unique population that may be at a higher risk of obesity due to the demanding nature of their studies and the associated stress [17]. Alzahrani et al. found that the prevalence of overweight and obesity among medical students in Saudi Arabia was 25.2% and 17.5%, respectively [18]. The study also identified several risk factors for obesity, including male sex, lack of physical activity, and consumption of fast food [18].

Despite the high prevalence of obesity among university students, few studies have investigated its prevalence and risk factors among medical students in Saudi Arabia. A study at Shaqra University showed that 8.8% of medical students were underweight, 46.5% had a normal BMI, 25.4% were overweight, and 19.3% were obese. These findings indicated that more than half of the participants had abnormal BMI values [19]. Interestingly, the study found that fast-food consumption did not have a significant effect on obesity, whereas regular exercise significantly reduced obesity [19]. However, to our knowledge, no study has investigated the prevalence and risk factors of obesity among medical students at Jazan University.

Therefore, this study aimed to determine the prevalence of overweight and obesity and the associated risk factors among medical students at Jazan University, Saudi Arabia. We hypothesize that the prevalence of overweight and obesity among medical students at Jazan University will be high and that various lifestyle, socioeconomic, and demographic factors will be associated with increased obesity risk in this population.

## 2. Methods

### 2.1. Study Design and Setting

This cross-sectional study was conducted among medical students at the Faculty of Medicine, Jazan University, Saudi Arabia, in the period from January to March 2024. The Faculty of Medicine at Jazan University is one of the leading medical schools in the region, with a diverse student body representing various socioeconomic and cultural backgrounds.

### 2.2. Study Population 

The study population consisted of medical students from the second to sixth academic years. The total number of medical students enrolled at Jazan University is around 900. Using an online sample-size calculator with a 95% confidence level, 5% margin of error, and an estimated 19% prevalence of obesity, the minimum required sample size was determined to be 188 participants. To account for potential non-response, we aimed to recruit an additional 20% of participants, resulting in a final sample size of 228 students. The study included students aged 18 or older from the College of Medicine at Jazan University while excluding those unwilling to participate.

### 2.3. Sampling Plan and Representation

The study employed a stratified random sampling technique to ensure proportional representation from each academic year. The student roster for each year was obtained from the faculty administration, and participants were randomly selected using a computer-generated random number list. This method ensures that all academic years are adequately represented and that the selected sample accurately reflects the demographics and characteristics of the broader population of medical students at Jazan University.

### 2.4. Data Collection

Data were obtained utilizing a validated questionnaire (Appendix A) derived from various prior studies. The questionnaire employed is a self-administered online tool comprising five distinct sections. The Section 1 encompasses socio-demographic information and incorporates variables such as age, gender, marital status, place of residence, academic achievement, socioeconomic status, and monthly income. The Section 2 captures measurements related to the body mass index (BMI) and waist circumference. Height and weight measurements were utilized to calculate the BMI, with BMI values falling below 18.5 kg/m^2^ indicating underweight, values between 18.5 and 24.9 kg/m^2^ indicating normal weight, values between 25 and 29.9 kg/m^2^ indicating overweight, and values equal to or exceeding 30 kg/m^2^ indicating obesity [19,20,21]. Waist circumference measurements are based on the guidelines set forth by the International Diabetes Federation, with waist circumference values ≤ 90 cm for males and ≤80 cm for females classified as waist 1, values between 90 and 102 cm for males and 80 and 88 cm for females classified as waist 2, and values exceeding 102 cm for males and exceeding 88 cm for females classified as waist 3. The Section 3 is specifically designed to gather data about physical activity and is adapted from the concise version of the International Physical Activity Questionnaire (IPAQ-short version) [22]. The Section 4 is structured to collect data concerning dietary patterns and is adapted from questions employed in the Food Frequency Questionnaire (FFQ) [23]. Lastly, the Section 5 entails the collection of data on comorbid conditions such as diabetes mellitus (DM), hypertension, and dyslipidemia. The final questionnaire consisted of the following sections:-Sociodemographic characteristics: age, gender, marital status, residence, academic year, socioeconomic status, and monthly income.-Physical activity: regular exercise, type, and intensity of physical activity.-Dietary habits: fruit and vegetable consumption, meal frequency, fast-food intake, and sugar-sweetened beverage consumption.-Comorbidities included hypertension, diabetes mellitus, and hyperlipidemia.-Medication use: corticosteroids, beta blockers, antidiabetics, antipsychotics, and antidepressants.-Family history: obesity, diabetes mellitus, and hypertension.-Lifestyle factors: smoking (cigarettes or shisha) and khat chewing.-Academic performance: current grade point average (GPA).

A pilot study was conducted among 10 medical students who were not included in the main study to test the questionnaire’s clarity, comprehensibility, and reliability. The internal consistency of the questionnaire was assessed using Cronbach’s alpha. Since the questions measured different constructs, we evaluated the internal consistency of individual subscales within the questionnaire where applicable. Cronbach’s alpha was calculated for each relevant subscale, with a value of 0.75 or higher considered satisfactory. Based on the pilot study results, minor modifications were made to the questionnaire to improve its clarity and flow.

### 2.5. Reasons for Non-Participation

Eligible participants who did not participate in the study cited various reasons, including lack of time due to academic commitments, personal reasons, and reluctance to share personal health information. Some students also mentioned concerns about the confidentiality of their data, despite assurances that all information would be anonymized and handled confidentially.

### 2.6. Handling Missing Data

Missing data were handled using multiple imputation methods to ensure the robustness of the results. Any incomplete responses in the questionnaires were imputed based on the observed data, maintaining the overall distribution and relationships within the dataset. Sensitivity analyses were conducted to assess the impact of missing data on the study findings, confirming that the results remained consistent.

### 2.7. Data Management and Analysis

Data were entered, cleaned, and analyzed using the Statistical Package for Social Sciences (SPSS) version 24.0. Descriptive statistics, including frequencies, percentages, means, and standard deviations, were used to summarize sociodemographic characteristics, anthropometric measurements, physical activity, dietary habits, comorbidities, medication use, family history, and lifestyle factors.

The prevalence of obesity was calculated as the proportion of participants with a BMI ≥ 30 kg/m^2^. BMI was calculated as the weight in kilograms divided by the height in meters squared (kg/m^2^). Participants were classified according to the World Health Organization’s BMI categories: underweight (BMI < 18.5 kg/m^2^), normal weight (BMI 18.5–24.9 kg/m^2^), overweight (BMI 25–29.9 kg/m^2^), class 1 obesity (BMI 30–34.9 kg/m^2^), class 2 obesity (BMI 35–39.9 kg/m^2^), and class 3 obesity (BMI ≥ 40 kg/m^2^). Waist circumference was measured by the participants themselves using a non-stretchable measuring tape. Participants were instructed to measure their waist at the midpoint between the lower margin of the last palpable rib and the top of the iliac crest, while standing and breathing normally. Measurements were recorded to the nearest 0.1 cm. Waist circumference was then categorized according to gender-specific cut-offs: normal (<80 cm for females and <90 cm for males), increased risk (80–88 cm for females and 90–102 cm for males), and substantially increased risk (>88 cm for females and >102 cm for males).

The chi-squared test was used to assess the association between categorical variables (sociodemographic factors, physical activity, dietary habits, comorbidities, medication use, family history, and lifestyle factors) and obesity. The significance level was set at *p* < 0.05. Binary logistic regression analysis was conducted to identify the risk factors for obesity. Variables with a *p*-value < 0.2 in the bivariate analysis were included in the multivariate logistic regression model. Adjusted odds ratios (AORs) and 95% confidence intervals (CIs) were calculated to determine the strength of the association between risk factors and obesity. Statistical significance was set at *p* < 0.05.

### 2.8. Ethical Considerations

The study protocol was approved by the Research Ethics Committee of the Faculty of Medicine of Jazan University. All participants provided informed consent before enrollment in the study. The study was conducted in accordance with the Declaration of Helsinki and the ethical principles for medical research involving human subjects. Participants’ confidentiality and anonymity were maintained throughout the study. Each participant was assigned a unique identification number, and personal information was not collected. Completed questionnaires and anthropometric data were stored securely with access restricted to the research team. Participants were informed of their right to withdraw from the study at any time without consequences.

## 3. Results

The sociodemographic characteristics and anthropometric measurements of the 228 university students are shown in Table 1. Most participants were between 18 and 22 years of age (139, 61.0%), female (126, 55.3%), unmarried (212, 93.0%), and evenly split between rural (114, 50.0%) and urban (114, 50.0%) areas. Most were in their second to fifth year of study, earned less than SAR 2000 monthly (180, 78.9%), and had grade-point averages above 4.5 (141, 61.8%). The average monthly family incomes were commonly between SAR 8000 and 14,999 (53, 23.2%) and between SAR 15,000 and 24,999 (61, 26.8%).

In terms of BMI, 54.4% (123) had a normal BMI, while 65.1% (148) of the students had normal or healthy waist circumferences. The mean weight was 63.39 ± 18.93 kg, and the mean height was 163.48 ± 9.78 cm. Underweight (39, 17.3%), overweight (30, 13.3%), and obesity (34, 15%) were also reported.

This study also investigated the exercise habits of the participants, as shown in Figure 1. The results revealed that most students (61%, *n* = 139) did not engage in regular exercise, while only 39% (*n* = 89) reported exercising regularly. Among the participants who reported exercising regularly (*n* = 89), this study further explored the duration of their weekly exercise. The results showed that 43.6% (*n* = 48) of these students exercised for less than two and a half hours per week, while 30% (*n* = 33) exercised for two and a half hours per week. Interestingly, 26.4% (*n* = 29) of the regularly exercising participants reported engaging in physical activity for more than two and a half hours per week.

The study also investigated the dietary habits of the participants, focusing on their consumption of fruits, vegetables, fast food, and sugar-sweetened soft drinks, as well as their daily meal frequency (Figure 2). When asked how often they consumed fruit per week, 189 out of 228 respondents (82.9%) said 3 or fewer days. Only 39 out of 228 (17.1%) participants reported eating fruits for more than 3 days per week. Regarding vegetable consumption, 134 out of 228 (58.8%) said they eat vegetables 3 or fewer days per week, while 94 out of 228 (41.2%) consumed them for more than 3 days. In terms of the number of meals eaten daily, 192 of 228 (84.2%) reported eating 3 or fewer meals per day, with only 36 of 228 (15.8%) eating more than 3 meals daily. When asked about weekly fast-food consumption, 132 out of 228 (57.9%) said they eat fast food 3 or fewer days per week, compared to 96 out of 228 (42.1%) who eat it more than 3 days. Finally, 155 out of 228 (68%) reported drinking sugar-sweetened soft drinks 3 or fewer days per week, while 73 out of 228 (32%) drank them on more than 3 days weekly.

A survey was conducted to collect data on health conditions, medication use, and family history as shown in Table 2. Regarding comorbidities, 8 (3.5%) reported having high blood pressure, 10 (4.4%) had diabetes, and 22 (9.6%) suffered from high blood-fat levels. In terms of medication use, 14 (6.1%) were taking corticosteroids, beta-blockers, antidiabetics, antipsychotics, or antidepressants. When asked about family history, 72 (31.6%) had a first-degree relative with obesity, 96 (42.1%) had a family history of diabetes, and 107 (46.9%) had relatives with high blood pressure. Regarding lifestyle factors, 14 (6.1%) currently smoked cigarettes or shisha, while only 3 (1.3%) chewed khat.

The results of the chi-square tests for association between categorical variables and obesity are presented in Table 3. Additionally, binary logistic regression analysis was conducted to identify risk factors for obesity. The collinearity diagnostic tests were performed, and the variance inflation factor (VIF) values for all independent variables were below 2, indicating no significant multicollinearity issues.

Table 3 presents the potential relationships among obesity, sociodemographic characteristics, and physical activity. The prevalence of obesity was 18.0% in 18–22 year olds, 11.0% in 22–26 year olds, and 28.6% in those over 26 years old (*p* = 0.247). By sex, 18.6% of males and 13.5% of females were obese (*p* = 0.290). Marital status showed little difference, with 12.5% of married and 16.0% of unmarried individuals being obese (*p* = 0.755). Regarding residence, 14.0% of the rural and 17.5% of the urban participants were obese (*p* = 0.468). Across academic levels, the obesity prevalence ranged from 10.5% in the 6th years to 20.7% in the 3rd years (*p* = 0.746). Family income was not significantly associated, with obesity prevalence highest in those earning < SAR 4000 (31.6%) and lowest in those earning SAR 15,000–24,999 (9.8%) (*p* = 0.273). Personal monthly income showed an 8.3% obesity prevalence in those earning > SAR 2000 compared to 17.8% in those earning less (*p* = 0.111). Finally, 14.6% of regular exercisers were obese versus 16.5% of non-exercisers (*p* = 0.695). None of the sociodemographic factors or physical activity levels showed a statistically significant association with obesity (all *p* > 0.05).

Table 4 analyzes the relationship between obesity and dietary habits among the 228 participants. Regarding weekly fruit consumption, 15.9% of those eating fruits ≤3 days/week were obese, compared to 15.4% of those eating fruits >3 days/week (*p* = 0.939). Regarding weekly vegetable intake, 16.4% of those eating vegetables ≤3 days/week were obese and 14.9% of those eating vegetables >3 days/week (*p* = 0.756). For the number of meals per day, obesity prevalence was 15.6% in those eating ≤3 meals/day and 16.7% in those eating >3 meals/day (*p* = 0.875). Analyzing weekly fast food consumption, 14.4% of those eating ≤3 days/week were obese, compared to 17.7% of those eating >3 days/week (*p* = 0.498). Regarding the weekly intake of sugary drinks, 16.1% of those drinking them ≤3 days/week were obese, as were 15.1% of those drinking them >3 days/week (*p* = 0.838). Finally, a family history of obesity showed no significant association, with 16.7% of those with a family history of obesity compared to 15.4% without (*p* = 0.805). This study found no statistically significant associations between obesity prevalence and any of the dietary habits examined, including fruit and vegetable consumption, meal frequency, fast-food intake, sugary drink intake, or family history (all *p*-values > 0.05).

Table 5 shows the potential associations between obesity and health conditions, medication use, and education level in 228 participants. Regarding comorbidities, a similar proportion of those with high blood pressure were obese (25.0%) compared to those without high blood pressure (15.5%), but the difference was not statistically significant (*p* = 0.615). Regarding diabetes, 30.0% of those with diabetes were obese versus 15.1% of those without diabetes (*p* = 0.197). High blood-fat levels showed a non-significant association, with 27.3% of those with high fat levels being obese compared to 14.6% without obesity (*p* = 0.129). Medication use had an intriguing pattern, with none (0%) of those on corticosteroids, beta-blockers, antidiabetics, antipsychotics, or antidepressants being obese, compared to 16.8% of non-users being obese (*p* = 0.134). Finally, education level measured by GPA demonstrated no significant association with obesity (*p* = 0.975).

Binary logistic regression analysis was conducted to identify sociodemographics, physical activity-related, dietary habits, and family history as potential risk factors for obesity among participants (Table 6). Compared to the reference group of 18–22-year-olds, the odds of obesity were lower in 22–26-year-olds (OR 0.422) but higher in those over 26 years (OR 1.915), although these age differences were not statistically significant (*p* > 0.05). Females had 50% lower odds of obesity than males (OR 0.500, *p* = 0.103). Unmarried participants had twice the odds of obesity versus married participants (OR 2.046, *p* = 0.445). Urban residence was associated with a non-significant 41.6% greater odds of obesity than rural residence (OR 1.416, *p* = 0.387). Across academic levels, the odds of obesity were elevated in the 3rd years (OR 1.715) and 5th years (OR 1.553) but lowered in the 4th years (OR 0.964) and 6th years (OR 0.995); however, none of these reached statistical significance (*p* > 0.05). Those with monthly family incomes of SAR 15,000–24,999 had significantly lower odds of obesity (OR 0.230, *p* = 0.045) than the lowest-income group. Regular exercisers had 31.7% greater odds of obesity than did non-exercisers (OR 1.317, *p* = 0.509). Compared to eating fruits ≤3 days/week, eating them >3 days/week was not associated with significantly greater odds of obesity (OR 1.008, *p* = 0.989). Eating vegetables >3 days/week was associated with lower odds of obesity versus eating them ≤3 days/week (OR 0.881), but this was not statistically significant (*p* = 0.758). Eating >3 meals per day did not significantly impact obesity odds compared to eating ≤3 meals per day (OR 1.081, *p* = 0.890). Those who ate fast food >3 days/week had 1.368 times higher odds of obesity compared to those eating it ≤3 days/week (*p* = 0.439). Drinking sugary drinks >3 days/week was associated with lower obesity odds than drinking ≤3 days/week (OR 0.810, *p* = 0.625). Having a family history of obesity did not significantly affect the odds of developing obesity (OR 0.899, *p* = 0.785). 

## 4. Discussion

The present study aimed to determine the prevalence of obesity and its associated risk factors among medical students at Jazan University, Saudi Arabia. Overall, the combined prevalence of overweight and obesity among the participants was 28.3%. A similar previous study conducted among medical students in Saudi Arabia at Shagra University by Ahmad et al. reported prevalence rates of 45% and 19% for overweight and obesity, respectively [19]. In the present study, the prevalence rates of overweight and obesity were 13.3 and 15%, respectively. Notably, the prevalence rates of overweight and obesity in our study are relatively lower than the findings in the Ahmad et al. study. These differences might be attributed to variations in sample size, geographical location, and cultural factors. However, both studies indicate a significant burden of excess weight among medical students in Saudi Arabia. Similarly, the prevalence of obesity in the present study (15%) is consistent with what was found among male university students in another Saudi Arabian study (15.7%) [8].

On the other hand, the prevalence of overweight and obesity in the present study was higher than that reported among university students in low- and middle-income countries (22%) [14]. 

Research conducted among university students in developing countries reveals a notable prevalence of overweight and obesity. In Africa, for instance, studies have reported rates of 10% in Nigeria to 59.4% in Egypt. In Asia, rates ranging from 2.9% to 30.1% in China and between 30 and 30% in Malaysia [14]. In Latin America, reported rates are between 12.4% and 16.7% in Colombia, while Mexico’s prevalence stands at 31.6%. The Middle East and Near East regions have also shown significant rates, with Oman reporting 28.2%, Kuwait reporting 42%, Iran reporting 12.4%, and Turkey ranging from 10% to 47.4% [14]. More recently, Younis et al. reported a combined prevalence of overweight and obesity of 65% among healthcare workers in the Gaza Strip [20].

The wide variations of the prevalence rates of overweight and obesity observed in the present study compared to similar previous studies may be attributed to various factors including the differences in the sample characteristics, methodological variations, increased awareness, and behavior change, and environmental factors can also contribute to the disparities in prevalence rates.

This study investigated the relationship between obesity and various sociodemographic factors, physical activity, dietary habits, comorbidities, medication use, and family history. Although no statistically significant associations were found between obesity and most of these factors, this study highlighted some important trends and patterns. Regarding physical activity, the study found that the majority of students (61%) did not engage in regular exercise. This finding is consistent with previous studies that have reported high rates of physical inactivity among university students in Saudi Arabia and other Arab countries [15,16]. Physical inactivity is a well-established risk factor of obesity and other chronic diseases [10,11]. Therefore, promoting regular physical activity among medical students should be a priority in obesity prevention and management.

The dietary habits of participants were also assessed. A high proportion of students consumed fruits (82.9%) and vegetables (58.8%) three or fewer days per week, and 84.2% ate three or fewer meals per day. Moreover, 42.1% of the participants reported consuming fast food more than three days per week. These dietary patterns are similar to those reported in other studies of university students in Saudi Arabia and the Middle East [8,16]. Unhealthy dietary habits, such as low fruit and vegetable intake, skipping meals, and frequent consumption of fast food, have been associated with an increased risk of obesity and related metabolic disorders [10,11].

Although the present study did not find significant associations between obesity and comorbidities, medication use, or family history, these factors should not be overlooked. Previous research has shown that obesity is associated with an increased risk of hypertension, diabetes, and dyslipidemia [24,25,26]. Additionally, certain medications, such as corticosteroids and antipsychotics, have been linked to weight gain and obesity [27,28]. Family history is also an important risk factor for obesity, as genetic and environmental factors can contribute to the development of this condition [29,30].

In the present study, a considerable proportion of participants, specifically 17.3%, fell under the category of underweight. This finding underscores the importance of addressing not only overweight and obesity but also the issue of insufficient weight, as it can also have adverse health implications.

Universities and colleges have the unique opportunity to promote healthy behaviors among students. Unhealthy lifestyles are common among college students, emphasizing the importance of implementing comprehensive health education programs [31]. Tackling obesity among medical and healthcare students is especially crucial, as they are the future healthcare providers and represent role models for their communities.

The present study has several strengths, including the use of a validated questionnaire and the inclusion of a diverse sample of medical students from different academic years. However, it has several limitations. First, the cross-sectional design does not allow for the establishment of causal relationships between obesity and the investigated risk factors. Second, the reliance on self-reported data for variables such as BMI, waist circumference, dietary habits, and physical activity may have introduced recall bias. Third, the sample was limited to medical students at a single university, which may limit the generalizability of the findings to other student populations. Additionally, the study did not account for potential confounding factors such as psychological stress and sleep patterns, which could also influence obesity. Future research should consider a longitudinal design and include a more diverse sample to validate and extend these findings.

On the other hand, this study’s findings have important implications for public health and medical education. The high prevalence rate of abnormal BMI (45.6%) and unhealthy lifestyle behaviors among medical students in the present study highlights the need for targeted interventions to promote healthy eating and regular physical activity in this population. Medical schools should integrate nutrition and physical activity education into their curricula and provide supportive environments that encourage healthy lifestyle choices [18,30]. Moreover, addressing the mental health and well-being of medical students is crucial, as psychological stress has been associated with an increased risk of obesity [32,33].

Future research should employ a longitudinal design to better understand the temporal relationships between obesity and its risk factors among medical students. Qualitative studies may also provide valuable insights into the barriers and facilitators of healthy lifestyle behaviors in this population. Additionally, researchers should evaluate the effectiveness of interventions designed to prevent and manage obesity in medical students.

## 5. Conclusions

This study found a high prevalence of obesity among medical students at Jazan University. Key findings include the lack of significant associations between obesity and most sociodemographic, physical activity, and dietary factors, except for lower odds of obesity in students with a higher family income. The high rates of physical inactivity and poor dietary habits underscore the need for targeted interventions to promote healthier lifestyles among medical students. Future research should focus on identifying effective strategies for obesity prevention and management in this population.

## Figures and Tables

**Figure 1 medicina-60-00940-f001:**
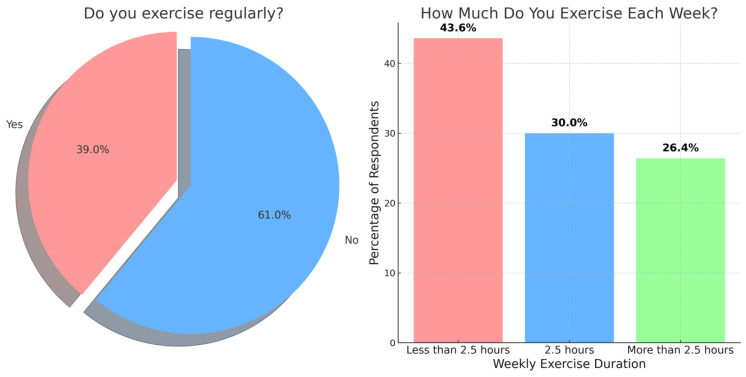
The exercise habits of the participants.

**Figure 2 medicina-60-00940-f002:**
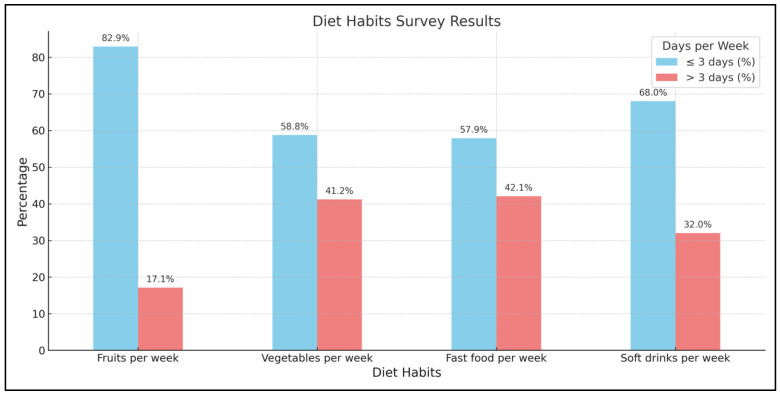
Dietary habits of the participants.

**Table 1 medicina-60-00940-t001:** Sociodemographic characteristics of the participants.

Sociodemographic Characteristics			*N*	%
Age group		18–22 years old	139	61.00%
		23–26 years old	82	36.00%
		Above 26 years	7	3.10%
Gender		Male	102	44.70%
		Female	126	55.30%
Marital status		Married	16	7.00%
		Unmarried	212	93.00%
Residence		Rural area	114	50.00%
		Urban Area	114	50.00%
Academic level		2nd year	68	29.80%
		3rd year	58	25.40%
		4th year	21	9.20%
		5th year	43	18.90%
		6th year	38	16.70%
Monthly family income (in riyals)		Less than 4000	19	8.30%
		4000–7999	25	11.00%
		8000–14,999	53	23.20%
		15,000–24,999	61	26.80%
		More than 25,000	70	30.70%
Monthly income of the student (in riyals)		Less than 2000	180	78.90%
		More than 2000	48	21.10%
What is your current GPA?		Less than 3	11	4.80%
		3–4.5	76	33.30%
		More than 4.5	141	61.80%
BMI		Less than 18.5	39	17.30%
		18.5–24.9	123	54.40%
		25–29.9	30	13.30%
		30–34.9	20	8.80%
		35–39.9	9	4.00%
		40 and above	5	2.20%
Waist circumference	Female	Less than 80 cm	94	41.20%
		80–88 cm	73	32.00%
		More than 88 cm	28	12.30%
	Male	Less than 90 cm	22	9.60%
		90–102 cm	5	2.20%
		More than 102 cm	6	2.60%
	Combined	Normal	116	50.80%
		<80 cm for females		
		<90 cm for males		
		Increased risk	78	34.20%
		80–88 cm for females		
		90–102 cm for males		
		Substantially increased	34	14.90%
		>88 cm for females		
		>102 cm for males		
Weight (kg); mean and SD		63.39 ± 18.93		
Height (in cm); mean and SD		163.48 ± 9.78		

**Table 2 medicina-60-00940-t002:** Comorbidities, medication, and family history.

Factors		*N*	%
Do you suffer from high blood pressure?	Yes	8	3.5%
	No	220	96.5%
Do you suffer from diabetes?	Yes	10	4.4%
	No	218	95.6%
Do you suffer from high blood-fat levels?	Yes	22	9.6%
	No	206	90.4%
Are you currently using a medication that belongs to any of the following drug classes? (Corticosteroids, B-blockers, antidiabetics, antipsychotics, antidepressants.)	Yes	14	6.1%
	No	214	93.9%
Do you have a family (first-degree relative) history of obesity?	Yes	72	31.6%
	No	156	68.4%
Do you have a family history (diabetes) of first-degree relatives?	Yes	96	42.1%
	No	132	57.9%
Do you have a family history (blood pressure) of first-degree relatives?	Yes	107	46.9%
	No	121	53.1%
Do you smoke (cigarettes or shisha)?	Yes	14	6.1%
	No	214	93.9%
Do you chew (use) khat?	Yes	3	1.3%
	No	225	98.7%

**Table 3 medicina-60-00940-t003:** Prevalence of obesity by sociodemographic factors and levels of physical activity.

Sociodemographic Factors		Obesity				*p*-Value
		Not Obese		Obese		
		*N*	*N*%	*N*	*N*%	
Age group	18–22 years old	114	82.0%	25	18.0%	0.247
	22–26 years old	73	89.0%	9	11.0%	
	Above 26 years	5	71.4%	2	28.6%	
Gender	Male	83	81.4%	19	18.6%	0.290
	Female	109	86.5%	17	13.5%	
Marital status	Married	14	87.5%	2	12.5%	0.755
	Unmarried	178	84.0%	34	16.0%	
Residence	Rural area	98	86.0%	16	14.0%	0.468
	Urban Area	94	82.5%	20	17.5%	
Academic level	2nd year	58	85.3%	10	14.7%	0.746
	3rd year	46	79.3%	12	20.7%	
	4th year	18	85.7%	3	14.3%	
	5th year	36	83.7%	7	16.3%	
	6th year	34	89.5%	4	10.5%	
Monthly family income (in riyals)	Less than 4000	13	68.4%	6	31.6%	0.273
	4000–7999	21	84.0%	4	16.0%	
	8000–14,999	44	83.0%	9	17.0%	
	15,000–24,999	55	90.2%	6	9.8%	
	More than 25,000	59	84.3%	11	15.7%	
Monthly income of the student (in riyals)	Less than 2000	148	82.2%	32	17.8%	0.111
	More than 2000	44	91.7%	4	8.3%	
Do you exercise regularly?	Yes	76	85.4%	13	14.6%	0.695
	No	116	83.5%	23	16.5%	

**Table 4 medicina-60-00940-t004:** Prevalence of obesity by dietary habits of the participant.

Dietary Habits		Obesity				*p*-Value
		Not Obese		Obese		
		*N*	*N*%	*N*	*N*%	
How often do you consume fruits per week?	Less than or equal to 3 days	159	84.1%	30	15.9%	0.939
	More than 3 days	33	84.6%	6	15.4%	
How often do you consume vegetables per week?	Less than or equal to 3 days	112	83.6%	22	16.4%	0.756
	More than 3 days	80	85.1%	14	14.9%	
How many meals do you eat a day?	Less than or equal to three days	162	84.4%	30	15.6%	0.875
	More than 3 days	30	83.3%	6	16.7%	
How many days do you usually eat fast food each week?	Less than or equal to 3 days	113	85.6%	19	14.4%	0.498
	More than 3 days	79	82.3%	17	17.7%	
How many days a week do you drink sugar-sweetened soft drinks?	Less than or equal to 3 days	130	83.9%	25	16.1%	0.838
	More than 3 days	62	84.9%	11	15.1%	
Do you have a family (first-degree relative) history of obesity?	Yes	60	83.3%	12	16.7%	0.805
	No	132	84.6%	24	15.4%	

**Table 5 medicina-60-00940-t005:** Prevalence of obesity by comorbidities, medications, and academic achievement.

Factors		Obesity				*p*-Value
		Not Obese		Obese		
		*N*	*N*%	*N*	*N*%	
Do you suffer from high blood pressure?	Yes	6	75.0%	2	25.0%	0.615
	No	186	84.5%	34	15.5%	
Do you suffer from diabetes?	Yes	7	70.0%	3	30.0%	0.197
	No	185	84.9%	33	15.1%	
Do you suffer from high blood-fat levels?	Yes	16	72.7%	6	27.3%	0.129
	No	176	85.4%	30	14.6%	
Are you currently using a medication that belongs to any of the following drug classes? (Corticosteroids, B-blockers, antidiabetics, antipsychotics, antidepressants.)	Yes	14	100.0%	0	0.0%	0.134
	No	178	83.2%	36	16.8%	
What is your current GPA ? *	Less than 3	9	81.8%	2	18.2%	0.975
	3–4.5	64	84.2%	12	15.8%	
	More than 4.5	119	84.4%	22	15.6%	

* GPA: grade point average, measured on a scale from 0 to 5.

**Table 6 medicina-60-00940-t006:** Assessing sociodemographic, physical activity-related, dietary, and familial risk factors for obesity using binary logistic regression.

Variable	B	S.E.	Wald	df	Sig.	Exp(B)
Age group	18–22 years (reference category)					
22–26 years	−0.863	0.637	1.834	1	0.176	0.422
>26 years	0.65	1.14	0.325	1	0.569	1.915
Gender	Male (reference Category)					
Female	−0.693	0.425	2.651	1	0.103	0.5
Marital status	Married (reference category)					
Unmarried	0.716	0.937	0.583	1	0.445	2.046
Residence	Rural area (reference category)					
Urban area	0.348	0.402	0.75	1	0.387	1.416
Academic level	2nd year (reference category)					
3rd year	0.539	0.514	1.101	1	0.294	1.715
4th year	−0.036	0.767	0.002	1	0.962	0.964
5th year	0.44	0.736	0.358	1	0.55	1.553
6th year	−0.005	0.874	0	1	0.996	0.995
Monthly family income (in riyals)	Less than SAR 4000 (reference category)					
SAR 4000–7999	−0.662	0.838	0.623	1	0.43	0.516
SAR 8000–14,999	−0.658	0.701	0.88	1	0.348	0.518
SAR 15,000–24,999	−1.469	0.733	4.019	1	0.045	0.23
More than SAR 25,000	−0.836	0.714	1.372	1	0.241	0.433
Monthly income of student	Less than SAR 2000 (reference category)					
More than SAR 2000	−0.858	0.598	2.057	1	0.152	0.424
Do you exercise regularly?	Yes (reference category)					
No	0.276	0.417	0.436	1	0.509	1.317
How often do you consume fruits per week?	Less than or equal to three days (reference category)					
More than 3 days	0.008	0.58	0	1	0.989	1.008
How often do you consume vegetables per week?	Less than or equal to three days (reference category)					
More than 3 days	−0.127	0.411	0.095	1	0.758	0.881
How many meals do you eat a day?	Less than or equal to three meals per day (reference category)					
More than 3 meals	0.078	0.563	0.019	1	0.89	1.081
How many days do you usually eat fast food each week?	Less than or equal to three days (reference category)					
More than 3 days	0.314	0.405	0.6	1	0.439	1.368
How many days a week do you drink sugar-sweetened soft drinks?	Less than or equal to three days (reference category)					
More than 3 days	−0.211	0.432	0.239	1	0.625	0.81
Do you have a family (first-degree relative) history of obesity?	Yes (reference category)					
No	−0.107	0.392	0.074	1	0.785	0.899

## Data Availability

The data presented in this study are available on request from the corresponding author due to restrictions related to subject privacy.

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
