# Peer review of "Overweight, Obesity, and Associated Risk Factors among Students at the Faculty of Medicine, Jazan University"

_medicina, 2024, doi:10.3390/medicina60060940_

Round 1
Reviewer 1 Report
Comments and Suggestions for Authors
Comments on the Quality of English LanguageAuthor Response
We would like to express our sincere gratitude for the review and feedback concerning our manuscript number 3010055 entitled "Prevalence of Overweight, Obesity, and Associated Risk Factors among Students at the Faculty of Medicine, Jazan University."
Your insightful suggestions no doubt will improve the quality and clarity of our work. We truly appreciate the time and effort you dedicated to thoroughly reviewing our research paper.
Kindly see our point-by-point responses to your comments and suggestions which are shown in the attached file.

Reviewer 2 Report
Comments and Suggestions for Authors
Thank you for the opportunity to review the work ID: medicina-3010055. This study aimed to determine the prevalence of overweight, obesity and its associated risk factors among medical students at Jazan University, Saudi Arabia.
Comments (Major revision):
Section Introduction:
In the Introduction section, a detailed description of previous studies on this topic is given, citing relevant references. The goal of the work is clearly defined.
Methods:
Authors mention the study population but reading this does not clearly provide who are the study population. A flow chart or any other visualization would greatly increase the clarity of this.
How were participants selected (sampling plan)? How well do they represent the study population?
State the reasons for not participating in this study.
Their no information about missing data and how the author handled the missing data.
Lines 112-136: Give references for questionnaires, eg. for the Food Frequency Questionnaire, etc.
Lines 171-174: This paper has no results for the statistical tests described in this text. Correct this.
State the results of the collinearity test.
Discussion Section:
Lines 325-331: Provide appropriate references for all studies cited.
Lines 377-380: Very scant discussion of the shortcomings of this study. Correct this.
Conclusion: Shorten this text, in order to avoid repeating the results, highlight the important results of the study.
Author Response
Reviewer 2 Comments and Responses
Dear Editors and Reviewers / Medicina Journal,
We would like to express our sincere gratitude for the review and feedback concerning our manuscript number 3010055 entitled 'Prevalence of Overweight Obesity and Associated Risk Factors among Students at the Faculty of Medicine Jazan University.' Your insightful suggestions will no doubt improve the quality and clarity of our work. We truly appreciate the time and effort you dedicated to thoroughly reviewing our research paper. Kindly see our point-by-point responses to your comments and suggestions (attached file)

Reviewer 3 Report
Comments and Suggestions for Authors
The objective of this cross-sectional study was to ascertain the prevalence of overweight, obesity, and associated risk factors among medical students at Jazan University in Saudi Arabia.
- Abstract: While well-organized, it is lengthy. Suggested revisions include removing background information and condensing results.
- Introduction: Add the research hypothesis at the conclusion of the introduction.
- Methodology: Clarify the method for calculating sample size. The population under study should be explicitly stated, resulting in 228 respondents. Specify inclusion and exclusion criteria for participants. Provide a reference for the validated questionnaire used in the study and mention the program utilized for the online questionnaire.
- Results: Ensure that all text and images, particularly in Picture 2, are legible. Adjust the font size if necessary. Additionally, avoid including unnecessary text such as "Chi-square of" in tables.
- Ensure that the literature is cited in accordance with the journal's guidelines.
These revisions will enhance the clarity and compliance of the manuscript with journal requirements.
Author Response

(The authors gave the same response as above.)

Round 2
Reviewer 2 Report
Comments and Suggestions for Authors
Thank you for the opportunity to re-review the manuscript ID: medicina-3010055. The authors responded correctly to all my comments and made appropriate corrections in the revised version of this paper.
I believe that the corrections made have contributed to the informativeness and transparency of this paper.
I thank the authors for all the responses to my comments and the effort they put into improving this paper.
Author Response
Dear Editors of Medicina Journal,
We would like to thank you for your timely efforts to improve our study manuscript entitled "Prevalence of Overweight, Obesity, and Associated Risk Factors among Students at the Faculty of Medicine, Jazan University."
Reviewer 3 Report
Comments and Suggestions for Authors
Thanks to the authors for correcting what was requested.
Author Response

(The authors gave the same response as above.)
